# A Novel Approach for Measurement of Free-Form Optical Elements with Digital Holographic Microscopy

**DOI:** 10.3390/mi13101719

**Published:** 2022-10-12

**Authors:** Xuhui Zhang, Chonglin Wu, Lei Chen, Gengliang Chen, Guoliang Zheng

**Affiliations:** 1Sino-German College of Intelligent Manufacturing, Shenzhen Technology University, Shenzhen 518118, China; 2School of Mechanical Engineering, Dongguan University of Technology, Dongguan 523830, China

**Keywords:** DHM, free-form measurement, thickness measurement, surface measurement

## Abstract

Free-form optical elements face significant challenges in high-precision measurement due to their high complexity and non-rotational symmetry. Digital holographic microscopy (DHM), as one of the methods for the measurement of free-form optical elements, has promising applications due to its ultra-high precision and non-destructive and fast characteristics. Therefore, we have designed a novel measurement method that combines transmission DHM and reflection DHM to obtain thickness information and surface information of elements to deduce the 3D structure. With this method, we completed the measurement of a free-form optical element. The DHM system we built has recorded holograms under 4× and 20× objectives and successfully recovered the 3D surface shape of the element. The measurements are consistent with the designed and manufactured parameters, demonstrating the unique advantages of DHM for measuring special types of optical elements.

## 1. Introduction

Free-form optical elements currently play an important role in numerous fields. An increasing number of optical systems are using them as key elements to improve the performance of optical systems. For example, free-form optical elements are used in vehicle and street lamps to increase the effective illumination range and improve energy efficiency [1,2,3,4], and in head-mounted displays [5,6,7,8,9] and micro-projectors [10,11,12,13,14] to reduce size and weight and improve imaging quality. However, due to their high complexity and non-rotational symmetry, high-precision measurement of free-form optical elements faces significant challenges, namely, how to further improve measurement accuracy, measurement efficiency and measurement versatility to achieve the high performance and low cost required for their areas of application. The current measurement methods of free-form optical elements can be divided into contact and non-contact methods [15]. Contact methods use a point-by-point scanning approach for measurement with an accuracy of microns in the vertical direction only, and have the risk of contaminating the sample, such as the CMM and profilometer methods [16]. The CMM method can measure free-form elements of arbitrary face shapes, but is slow and accurate to the order no smaller than microns. The profilometer method is more accurate than the CMM method, but can currently only measure off-axis aspheric free-form surfaces. The high demands of polished optical free-form elements on measurement accuracy and range make contact methods no longer adequate. Since the 1990s, non-contact methods have received enormous attention due to their potential for full-field non-destructive measurement, with representative methods such as Shack–Hartmann wavefront detection, structured light 3D measurement and interferometry. The dynamic measurement range of the Shack–Hartmann wavefront detection method is greater than that of interferometry, but the lateral measurement resolution is not high. Structured light 3D measurement is mainly used to measure aspheric and a small number of off-axis aspheric surface shapes, with measurement accuracy mostly in the micron range. Interferometry, as one of the most accurate detection methods currently available, has been unanimously recognized in the field of detection of optical planes, spherical surfaces and even aspheric surfaces [17]. Moreover, DHM, an ultra-high precision interferometry method, has achieved sub-nanometer scale axial resolution [18,19,20], which enables fast and contamination-free detection and has wide potential in high precision measurement.

DHM is a quantitative optical imaging technique [21,22,23] that is able to capture the complex wavefront (amplitude and phase) of the light interacting with samples. Capturing the wavefront is performed by recording the spatial interference pattern of the beam that interacts with the sample (i.e., object beam) and a mutually coherent reference beam using a digital camera. Typically, off-line DHM is a Mach–Zender Interferometer [24] with one beam carrying the sample information acting as the object beam, and another as reference beam interfering with the object beam. There are two types of off-line DHMs, transmission and reflection. With the sample penetrated by laser, the thickness/phase information is carried to the camera. The reference beam of laser can interfere with this beam, and interference fringes are captured by the camera. Phase information is then restored by phase wrapping of the Fourier spectrum of the images. Unlike transmission DHM, the laser beam reflected by the sample instead of the penetrated beam is captured by the camera in reflection DHM [25].

In this paper, we have demonstrated the unique advantages of DHM for measuring special types of optical elements by analyzing the measurement results of a free-form optical element. We designed a novel method of measuring such an element, which obtains thickness and surface information by transmission and reflection of the DHM separately. By combining the thickness and surface information to deduce the structure of the element, the results are consistent with the designed and manufactured parameters.

## 2. Materials and Methods

The optical element under measurement is illustrated in Figure 1. The element has a concave bottom surface and microlens arrays at the top surface with overall external dimensions of 40mm × 40mm × 1.5mm. The key dimensions are the radius of curvature, the aperture diameter, the vector height of the microlens center, and the overall radius of curvature of the upper and lower surfaces, the respective values of which are r=900 μm, 2a=241.5 μm, Sag=10.9 μm, R1 =31.8 mm, and R2 =33.3 m. Light scattering is severe under measurement with white light interferometer [26] leading to inaccurate results. Therefore, we proposed a digital holographic interferometric method for measuring such an element, which combines transmission and reflection DHM to achieve a fully structured measurement.

The schematic diagrams and experiment setups of the DHM are shown in Figure 2. In transmission DHM (Figure 2a,c), the laser (Cobolt-08-NLD) with the wavelength of 632.8 nm and linewidth of 1 pm provides a superior coherent light to generate high contrast interferences. It is separated into two beams after passing through the polarization beam splitter (PBS). One beam passing through the microscope objective (MO, Olympus UPLFLN 4×, 20×) after transmitting the sample formed an enlarged object wave. The other beam is a planar reference wave. These two beams, with a small difference of incidence angle, interfere with each other after being reflected/transmitted by the non-polarization beam splitter (BS). The hologram is recorded by the CCD sensor (MERCURY, MER-504-10GM, 8-bit) and then sent to the computer for processing. The reference wave transmitted through the PBS is horizontally polarized, while the reflected object wave is vertically polarized. To achieve interference, the half-wave plate (P) is inserted to adjust the polarization of reference wave to the vertical direction. A condenser L1 (focal length f = 175 mm) is placed before MO to converge the laser beam so as to illuminate the sample. The reflection DHM (Figure 2b,d) with the same laser (632.8 nm, Cobolt-08-NLD) is separated into two beams after the PBS, and the object wave is formed only after the reflection of the light wave is illuminated on the sample. Three lenses, two L1 (f = 175 mm) and L2 (f = 200 mm), are placed in the setup to adjust the interferences.

Reference beam R(x,y) and the objective beam O(x,y) are coherent on the CCD recording surface to form a hologram. The beam intensity distribution of a hologram I(x,y) can be expressed as:(1)$$I\left(x,y\right)={\left|R\left(x,y\right)\right|}^{2}+{\left|O\left(x,y\right)\right|}^{2}+O\left(x,y\right)R^{*}\left(x,y\right)+O^{*}\left(x,y\right)R\left(x,y\right),$$
where * defines the complex conjugate.

The first two items in Equation (1) are the intensity distributions of the reference beam and the objective beam, respectively, which are only related to the amplitude of the beam and constitute the zero-level image. The other two items contain the amplitude and phase information of the objective beam, which are the positive level image and the negative level image, respectively.

The CCD image element size used in the number of pixels is 2048 × 2048, also giving the 8-bit gray level output. The MO magnification is 4×, the numerical aperture is 0.13, the focal length is 45.00 mm, and the theoretical limit resolution is 2.58 μm. In order to ensure that the zero-level image and the positive and negative first-level images do not overlap the spectrum, Nyquist’s sampling theorem needs to be satisfied, and the angle between the objective beam and the reference beam needs be satisfied with the following relationship:(2)$$\theta \leq \theta_{max}=\sin^{-1}\left(\frac{\lambda }{2\Delta x}\right),$$
where λ is the wavelength of light, and Δx is the CCD image element size.

Digital holographic reproduction is a computer simulation of the diffraction process that uses the phase information of the hologram to produce a 3D image. In our experiments, the digital holographic reproduction algorithm we use is the angular spectrum diffraction algorithm. The computer illuminates the hologram with the generated digital reference beam C(x,y), i.e., $C\left(x,y\right)\cdot O\left(x,y\right)R^{*}\left(x,y\right)$. According to the scalar diffraction theory, the angular spectrum diffraction equation is expressed as:(3)$$U\left(x_{i},y_{i}\right)=F^{-1}\left\{F\left[C\left(x,y\right)\cdot O\left(x,y\right)R^{*}\left(x,y\right)\right]\cdot H\left(f_{x},f_{y}\right)\right\},$$
where $U\left(x_{i},y_{i}\right)$ is the reproduction of the complex amplitude of the light wave of the object; F and $F^{-1}$ denote Fourier transform and Fourier inverse transform of the image, respectively; $f_{x}$,$f_{y}$ are the frequency domain coordinates; and $f_{x}=x/L_{x}$, $f_{y}=\frac{y}{L_{y}};$ and $H\left(f_{x},f_{y}\right)$ is the transfer function for angular spectral diffraction, given by:(4)$$H\left(f_{x},f_{y}\right)=\exp\left[\mathrm{jkd}\sqrt{1-{\left({\lambda f}_{x}\right)}^{2}-{\left({\lambda f}_{y}\right)}^{2}}\right],$$
where $j=\sqrt{-1};\;\lambda$ is the wavelength of light; k is the number of light waves; and $k=\frac{2\pi }{\lambda }$, d is the reproduction distance.

From Equation (3), it can be shown that only one forward and one inverse fast Fourier transform are required for holographic reproduction using the angular spectrum method, eliminating the scale factor on the holographic and image planes, so that the reproduction surface image element size is equal to the CCD image element size [27], i.e., ${\Delta x}_{i}$ = Δx,$\;{\Delta y}_{i}$ = Δy. Since the CCD recording plane is the imaging plane, the recording distance of the digital hologram is zero.

In wavefront reproduction, there are aberrations in the optical aberrations of the objective beam itself, as well as aberrations in the reconstructed phase obtained by means of the analogue digital reference beam. Such phase aberrations are superimposed on the objective phase, causing severe distortion in the reproduced phase distribution. The inclined aberrations are introduced by the inclined incident reference beam and are related to the angle of incidence. Secondary phase aberrations are introduced by the MO and are related to the objective distance, MO parameters and CCD position.

In our experiments, we have chosen the digital mask method to eliminate phase aberrations, i.e., the transformation of primary and secondary phase aberrations due to physical conditions, or the elimination of higher phase aberrations by the calculation of equations and the inverse programming of computer simulations.

## 3. Results and Discussion

The information about the thickness or surface of the object is stored in the hologram by recording the interference fringes between either the object light generated by the transmission or the reflection of DHM from the object path, and the reference light wave through a CCD with high resolution (2048 × 2048, 8 bits) on the other, which is eventually stored in a computer. Under the 4× objective, Figure 3a,b shows the original holograms recorded by the experimental setups shown in Figure 2a,b, respectively. The holographic interference fringes with high contrast value are clearly visible when the central part of the hologram is magnified as indicated in the upper-right corner of each picture.

In order to obtain higher quality holograms, spectral filtering is required to eliminate zero-level image and negative-level false image, i.e., to simulate a suitable digital filtering window on the spectrum of the original hologram, retaining only the spectrum of the positive-level real image, and then performing an inverse Fourier transform to obtain the new hologram. Figure 4a,b respectively show the Fourier transform in Figure 3c,d, and Figure 4c,d respectively show the spectral filtering of Figure 4a,b. Digital holographic reproduction is a computer simulation of the diffraction process that uses the phase information of the hologram to produce a 3D image. Figure 4e,f respectively show the phase maps generated after digital reconstruction of Figure 4a,b.

In wavefront reproduction, we eliminate phase aberrations by computing equations and inverse programming of computer simulations, which are then unfolded into a continuous phase distribution using a phase unwrapping algorithm. The refractive index (1.49) of the element is also considered for thickness calculation, resulting in the results shown in Figure 5. Figure 5a shows the surface profile information obtained from Figure 4b, and Figure 5e shows the thickness information obtained from Figure 4f. By combining the two, it is possible to fit the structure of the device, as in Figure 5c.

To verify the stability of MLA manufacturing processes, we made one measurement in the center and four at the edge of the sample. The results of the measurements are shown in Table 1. We deduced from the measurements that the average radius of curvature, aperture diameter, and vector height of the microlens center are r=899.4 μm, 2a=242 μm, and Sag=10.80 μm, respectively; and the overall radius of curvature of the upper and lower surfaces are R1 =31.47 mm and R2 =32.97 mm, respectively.

Optical elements are subject to manufacturing processes and heat dissipation during processing, resulting in deviations between actual and design dimensions. Taking into account manufacturing errors and measurement errors, the actual dimensions of the qualified parts are required to be within a certain fluctuation range of the design dimensions. By analyzing the reconstructed data, we determined that the dimensions obtained by DHM are consistent with the designed and manufactured parameters.

We replaced the objective to 20× in transmission DHM for an enlarged sample measurement to evaluate the surface roughness of the sample. The recorded hologram is illustrated in Figure 6a. We then used the same algorithm to reconstruct the thickness information of the sample and the result is shown in Figure 6b.

Roughness evaluation was carried out in horizontal (x) and vertical (y) directions of one microlens located in the center. The designed/measured sample curves and residue of these two values are shown in Figure 7. The measurement is well fitted with designed curves in both directions. Residues are shown at the bottom of Figure 7 and are approximately 400 nm which indicates the manufacturing process is well controlled.

## 4. Conclusions

The measurement method in this paper integrates holograms of transmitted DHM and reflected DHM and is an ingenious fusion of methods for reconstructing the shape of micro and nano elements by interferometric phasing. The object of measurement is an optical element with free-form surface features. We have obtained measured values for the key parameters and roughness of the element. Comparing the theoretical and measured values, we demonstrate that the element is consistent with the designed and manufactured structure. The measurement methods are non-contact, label-free, high resolution, real-time observation and off-line 3D reconstruction. This system can be applied to microscopic optical element characterization, MEMS measurements, life sciences and other fields. The high-precision measurement of free-form optical components with low scattering can also be achieved.

## Figures and Tables

**Figure 1 micromachines-13-01719-f001:**
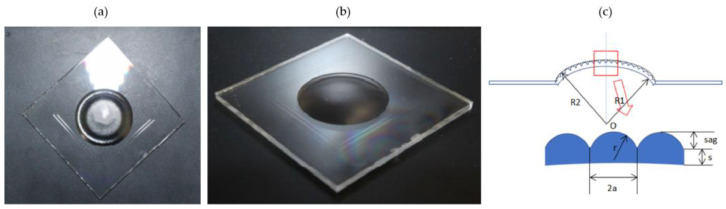
(**a**,**b**) The structure of the sample at different angles, MLAs on the top of the surface. (**c**) The designed structure of the sample, the MLA parameters shown in the zoomed image.

**Figure 2 micromachines-13-01719-f002:**
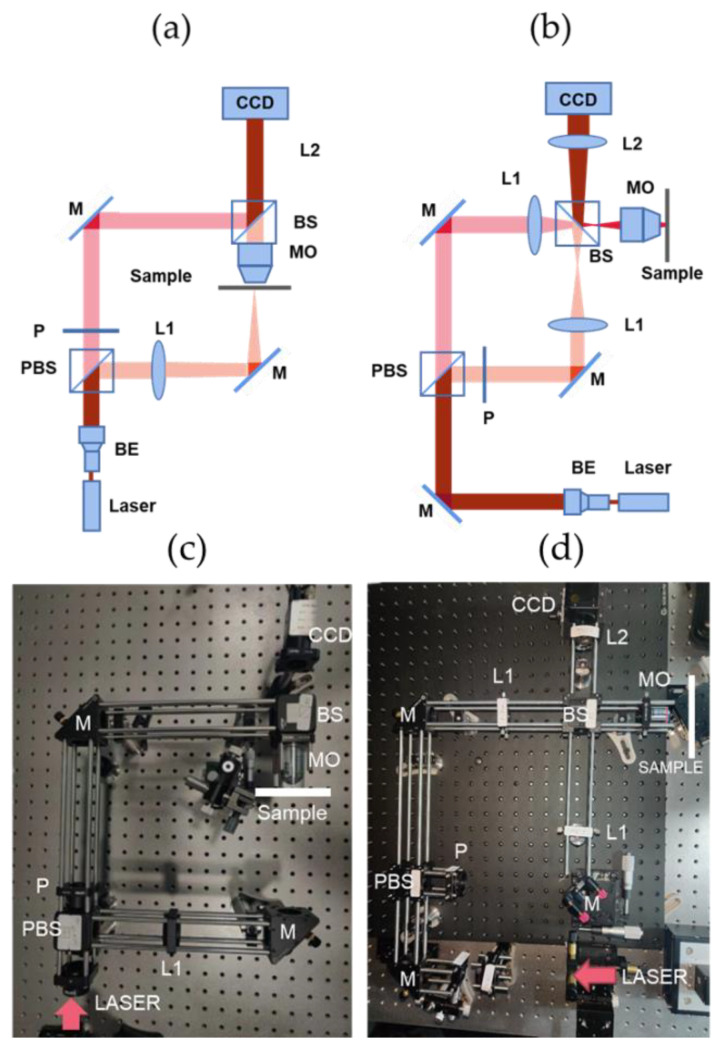
Schematic diagrams of DHM. (**a**) Transmission DHM. (**b**) Reflection DHM. The experiment setups of (**a**,**b**) are illustrated in (**c**,**d**), respectively. M: mirror, L1, L2: condenser, PBS: polarization beam splitter, BS: beam splitter, P: half-wave plate, MO: microscope objective, BE: beam expander.

**Figure 3 micromachines-13-01719-f003:**
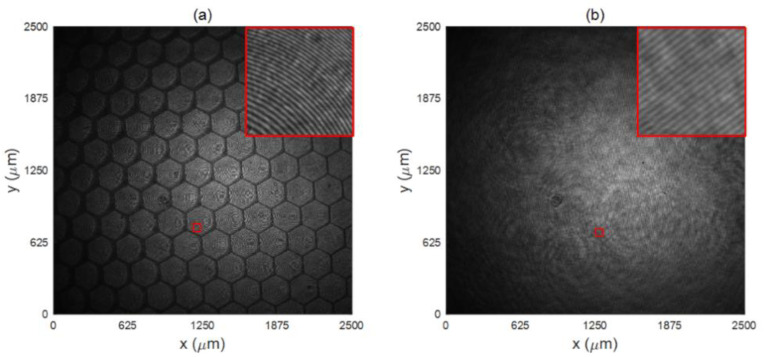
Digital hologram recorded by the CCD camera: (**a**) original hologram and magnified central part obtained by transmission DHM; (**b**) original hologram and magnified central part obtained by reflection DHM.

**Figure 4 micromachines-13-01719-f004:**
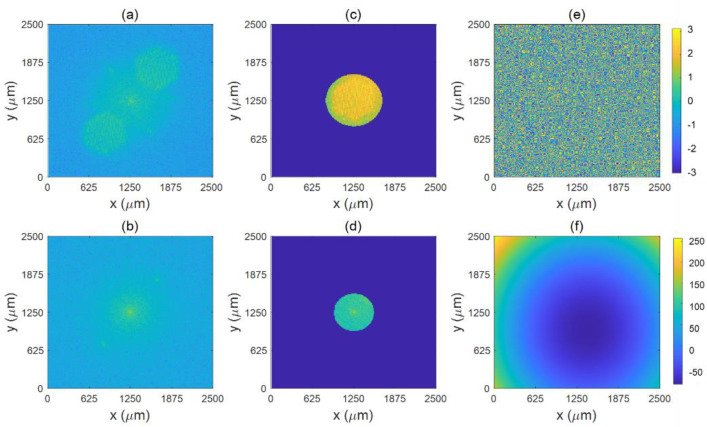
(**a**,**b**) The hologram doing the Fourier transform. (**c**,**d**) The frequency filtering done on (**a**,**b**), respectively. (**e**,**f**) The intensity map corresponding to the phase map.

**Figure 5 micromachines-13-01719-f005:**
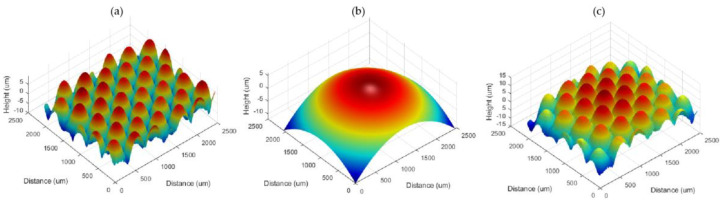
(**a**) Thickness information reproduced in the hologram is obtained from the transmission DHM. (**b**) Bottom surface information reproduced in the hologram is obtained from the reflection DHM. (**c**) The sample structure deduced from combination of bottom surface and thickness information.

**Figure 6 micromachines-13-01719-f006:**
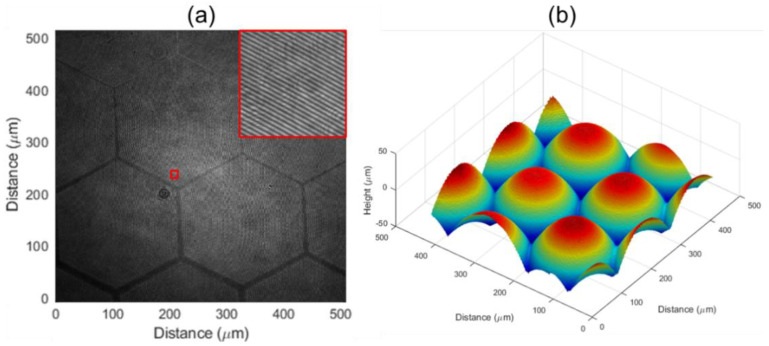
(**a**) Digital hologram recorded by transmission DHM and magnified central part. (**b**) Thickness information reproduced from (**a**).

**Figure 7 micromachines-13-01719-f007:**
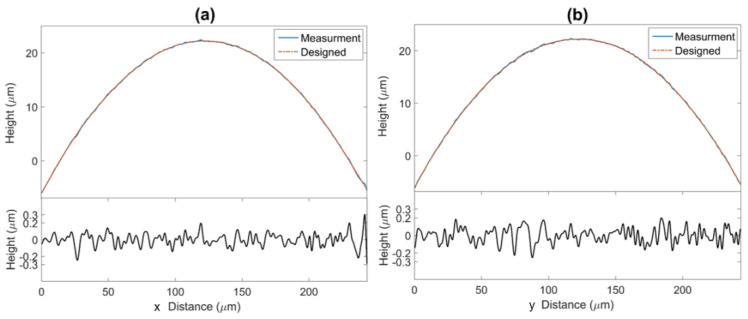
Sample curves and roughness measurements in the vertical and horizontal directions of the microlens. Measurement (blue line) and designed (red dashed line) height is shown at the top of each picture, the residue is illustrated at the bottom. (**a**) Horizontal direction, (**b**) Vertical direction.

**Table 1 micromachines-13-01719-t001:** Key parameter measurement results.

Parameters	Theoretical Value (Tolerance)	Average	Value of 5 Points
2a	241.5 μm (±5.0 μm)	242±0.20 μm	241.96 μm, 242.20 μm, 241.97 μm, 241.86 μm, 242.01 μm
r	900.0 μm (±2.5 μm)	899.4±1.50 μm	900.90 μm, 900.00 μm, 898.26 μm, 899.86 μm, 897.98 μm
R1	31.80 mm (±0.50 μm)	31.47±0.06 μm	31.51 μm, 31.41 μm, 31.47 μm, 31.44 μm, 31.52 μm
R2	33.30 mm (±0.50 μm)	32.97±0.06 μm	32.96 μm, 32.98 μm, 32.91 μm, 33.02μm, 32.98 μm
Sag	10.90 μm (±0.20 μm)	10.80±0.01 μm	10.79 μm, 10.80 μm, 10.79 μm, 10.81 μm, 10.81 μm

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
