# Peer review of "A Novel Approach for Measurement of Free-Form Optical Elements with Digital Holographic Microscopy"

_micromachines, 2022, doi:10.3390/mi13101719_

Round 1
Reviewer 1 Report
The measurement of free-form optical components face offers both a challenges and a scientific implication. In the article the authors provided a method which have ultra-high precision, non-destructive and fast characteristics. It has combines two DHM one for thickness the other for surface information measure. Although the method has been reported before, but the new contributions of the article would be recognized.
The following points need to be modified before the manuscript be accepted
1. English grammar needs improvement in the article.
2. Enhancing the description of introduction of holographic recording from lines 93 to 107.
3. The experimental and simulation results are well presented in the paper, but the theoretical part is weaker.
4. The information of the experimental equipment (such as CCD) and physical photos of the light path are missing.
5. The problem with the layout of the manuscript (lines 136)
Reviewer 2 Report
The manuscript presents a measurement method that combines transmission DHM and reflection DHM to obtain thickness information and surface information of free-form optical components to deduce the 3D structure. Experimental tests were performed and the measured results were compared with the designed parameters.
The results are interesting and could be important in the applications stated. I request the authors to address the following minor suggestions before the manuscript can be accepted for publication.
1、It is mentioned in the manuscript that the method can theoretically be applied to any free-form optical component measurement. What is the basis for the appropriateness of this statement.
2、Compare the effect of measuring free-form surfaces with traditional methods to show the unique advantages of this DHM method and how much the accuracy is improved.
3、It is recommended to add a color bar to mark the value of the color in Figure 4.
4、It is mentioned in the manuscript that the transmission DHM and reflection DHM use different working wavelengths, 632.8 mm emitted from He-Ne laser in transmission DHM and laser of 633 mm in reflection DHM. The unit of wavelength should be nm. Why do they use different wavelengths, and whether different wavelengths affect the accuracy?
5、Analyze the error between the measured results and the theoretical parameters and the causes of the error.
